# Antibacterial Activity against Four Fish Pathogenic Bacteria of Twelve Microalgae Species Isolated from Lagoons in Western Greece

**DOI:** 10.3390/microorganisms11061396

**Published:** 2023-05-25

**Authors:** Chrysa Androutsopoulou, Pavlos Makridis

**Affiliations:** Department of Biology, University of Patras, 26504 Patras, Greece

**Keywords:** antibacterial activity, fish pathogens, aquaculture, biological control

## Abstract

Microalgae may produce a range of high-value bioactive substances, making them a promising resource for various applications. In this study, the antibacterial activity of twelve microalgae species isolated from lagoons in western Greece was examined against four fish pathogenic bacteria (*Vibrio anguillarum*, *Aeromonas veronii*, *Vibrio alginolyticus*, and *Vibrio harveyi*). Two experimental approaches were used to evaluate the inhibitory effect of microalgae on pathogenic bacteria. The first approach used bacteria-free microalgae cultures, whereas the second approach used filter-sterilized supernatant from centrifuged microalgae cultures. The results demonstrated that all microalgae had inhibitory effects against pathogenic bacteria in the first approach, particularly 4 days after inoculation, where *Asteromonas gracilis* and *Tetraselmis* sp. (red var., Pappas) exhibited the highest inhibitory activity, reducing bacterial growth by 1 to 3 log units. In the second approach, *Tetraselmis* sp. (red var., Pappas) showed significant inhibition against *V. alginolyticus* between 4 and 25 h after inoculation. Moreover, all tested cyanobacteria exhibited inhibitory activity against *V. alginolyticus* between 21 and 48 h after inoculation. Statistical analysis was performed using the independent samples *t*-test. These findings suggested that microalgae produce compounds with antibacterial activity, which could be useful in aquaculture.

## 1. Introduction 

Microalgae are a diverse group of unicellular photosynthetic organisms [1], which can be found in various aquatic habitats, including freshwater, brackish, and marine environments. They form symbiotic relationships with a wide variety of other organisms, ranging from fungi to zooxanthellae [2]. Microalgae have been extensively studied for their potential as a source of bioactive compounds, such as fatty acids, phycobiliproteins, chlorophylls, carotenoids, and vitamins, which have various applications in the food, pharmaceutical, and cosmetic industries [3]. The lagoons of Messolonghi-Etoliko and the adjacent salt pans of Aspri and Tourlida in western Greece are unique ecosystems that support a rich diversity of microalgae. These habitats are characterized by extreme environmental conditions, such as high salinity, temperature, and light intensity in the summer, and low temperature and salinity in the winter. Microalgae species that live in these habitats have adapted to these changing environmental conditions and have developed various strategies to survive and thrive. For example, these microalgae have evolved to live in waters with a high organic and microbial load, so they may produce antioxidant or antimicrobial substances that influence their microenvironment and provide a competitive advantage [4].

In aquaculture rearing systems, fish or invertebrate populations are stocked at population densities much higher than in nature, and the spreading of disease is therefore much easier. Increased stress on the animals may depress the capacity of their immune system and make them more vulnerable to disease. Bacterial infections are a significant problem in aquaculture, as they can lead to high mortality rates in cultured populations of fish and invertebrates [5]. The misuse of antibiotics in aquaculture, such as oxytetracycline, florfenicol, enrofloxacin, and erythromycin [6], has led to the appearance of antibiotic-resistant bacterial strains [7,8]. These resistant strains may have a negative impact on both fish and human health [9] as they reduce the effectiveness of antimicrobial treatments in aquaculture and the development of antibiotic resistance. These resistant strains pose a significant threat to the environment, as they can persist in aquatic ecosystems, transferring their resistance genes to environmental bacteria and eventually to human pathogens [10,11]. Antibiotic-resistant bacteria can also deteriorate the environment by altering the natural balance of microbial communities in aquatic ecosystems. This can lead to a shift in the composition and diversity of these communities, which can have cascading effects on the food web and nutrient cycling [12]. In addition, the use of antibiotics in aquaculture can result in the accumulation of these compounds in the environment, which can have toxic effects on non-target organisms and influence the quality of water resources [13]. Therefore, it is crucial to limit the use of antibiotics in aquaculture and explore alternative strategies for disease control that minimize the development of antibiotic resistance and reduce the impact of aquaculture on the environment. A reduced use of antibiotics in aquaculture will improve consumers’ perception of the aquaculture industry.

Fish pathogens such as *Vibrio anguillarum*, which cause disease in marine fish, bivalves, and crustaceans, and *Aeromonas veronii*, which causes disease in freshwater and marine fish worldwide [14,15], result in significant economic losses in the aquaculture industry [16,17]. Other harmful bacteria such as *Vibrio alginolyticus* and *Vibrio harveyi* cause eye damage/blindness, gastroenteritis, muscle necrosis, skin ulcers, and tail rot disease [18,19,20,21]. 

Microalgae have been used in aquaculture as live food for various stages of bivalves, such as oysters, scallops, clams, and mussels, as well as for the rearing of marine fish larvae [22,23]. Microalgae are used for the production and enrichment of rotifers and can be added to fish tanks in the “green water technique” during the rearing of fish larvae. The use of microalgae as a source of antimicrobial compounds has gained increasing attention in recent years due to the emergence of antibiotic-resistant bacteria. The production of antimicrobial compounds by microalgae is a natural defense mechanism against microorganisms in their environment. Antibacterial activity in microalgae cultures can be caused either by bacteria associated with microalgae cultures [24] or by antibacterial substances produced by the microalgae cells [25]. Several studies have reported the antibacterial activity of microalgae against a range of pathogenic bacteria. Both eukaryotic microalgae and cyanobacteria have shown antifungal, antibacterial, and antiviral activity against a wide range of microorganisms. For example, cultures of *Tetraselmis* sp. have demonstrated antimicrobial activity and are commonly used in aquaculture; the antibacterial activity of *Tetraselmis* sp. hexane extracts was demonstrated in the case of *Staphylococcus aureus* [26]. Similarly, *Nephroselmis* sp. has shown antimicrobial and antioxidant properties due to its high carotenoid content [27,28], and hexane extracts of *Dunaliella salina* have shown antibacterial activity against *Bacillus subtilis* (BS), *Pectobacterium carotovorum* subsp. carotovorum (PCC), and *P. syringae* pv. tomato [29]. Petroleum ether, hexane, and ethanolic extracts of *D. salina* have shown antibacterial activity against *S. aureus* ATCC 25923 and *Εscherichia coli* ATCC 11775 [30]. Cyanobacteria have also shown antibacterial activity against multidrug-resistant (MDR) pathogenic bacteria and fish pathogens [31], as they produce a variety of secondary metabolites, organic compounds that are produced, and are not directly involved in the growth, development, or reproduction of the organism, that have shown antimicrobial activity against both Gram-positive and Gram-negative bacteria. Some examples of these secondary metabolites include phenazines, cyclic peptides, and lipopeptides [32,33,34,35]. Cyanobacteria in general inhibited fish pathogens such as Gram-negative *A. hydrophila* [36], and specifically, *Anabaena* sp. has also inhibited Gram-positive bacteria [37]. Finally, *Amphidinium carterae* produces a variety of secondary metabolites with potent anticancer, antifungal, and hemolytic activity, making it a potential source of new drugs [38]. Its antibacterial activity has been demonstrated against *S. aureus*, *Enterococcus faecalis*, *E. coli* and *Pseudomonas aeruginosa* [39], *Micrococcus*, *Aeromonas*, and *Vibrio* species [40]. The antibacterial activity of microalgae is often due to a variety of secondary bioactive metabolites. There are many chemically unique metabolites with different biological activity among microalgae species, and some of the antimicrobial activities of microalgae and cyanobacteria could be related to unsaturated fatty acids, such as eicosapentaenoic acid (EPA), hexadecatrienoic acid, and palmitoleic acid [41]. In addition, microalgae may produce oligopeptides or proteins with antibacterial activity, which bind to both polar and non-polar sites in bacterial cytoplasmic membranes, inhibiting cellular processes and cell division [42]. Antimicrobial peptides have been targeted as potential alternatives to antibiotics due to their broad antibacterial spectrum [43]. Finally, sulfated polysaccharide compounds could be involved in the antimicrobial activity of microalgae [41]. Cyanobacteria’s antibacterial activity has been specifically linked to a range of compounds including alkaloids, fatty acids, indoles, macrolides, peptides, phenols, pigments, and terpenes [44].

Biological control in aquaculture involves the use of live organisms to control the spreading of pathogens in a culture system. Several approaches have been suggested such as probiotic bacteria or yeast, bacteriophages, microalgae or macroalgae [45,46]. In this study, we aimed to investigate the potential of microalgae as a source of antibacterial agents for biological control in aquaculture. To achieve this goal, twelve microalgae species were isolated from the lagoons of Messolonghi-Etoliko and the adjacent salt pans of Aspri and Tourlida in western Greece. These microalgae were identified using a molecular approach, and their chemical composition was described [47]. These species were selected based on their prevalence in the lagoons and their potential for use in biotechnology applications. We then evaluated their antibacterial activity against four Gram-negative fish pathogens: *V. anguillarum*, *A. veronii*, *V. alginolyticus*, and *V. harveyi*, in two series of in vitro experiments.

## 2. Materials and Methods

### 2.1. Microorganisms and Growth Conditions

#### 2.1.1. Microalgae Cultures 

The microalgae used for the experiments were isolated from lagoons in western Greece [47]. These microalgae comprised eight chlorophytes: *Tetraselmis* sp. (red var.), *Tetraselmis* sp. (Red var., Pappas), *Tetraselmis* sp. (Red var., Kotichi), *Tetraselmis* sp. (Palmella), *Tetraselmis marina* (var. Messolonghi), *Nephroselmis* sp., *D. salina*, and *Asteromonas gracilis;* as well as three cyanobacteria: *Phormidium* sp., *Anabaena* sp., and *Cyanothece* sp.; and one dinoflagellate, *A. carterae*. The cultures were grown in sterile seawater in which Walne’s growth medium had been added [48] and kept in flasks under continuous light (9.25 × 10^−5^ mol × m^−2^ × s^−1^) at 22 °C. 

#### 2.1.2. Fish Pathogens

Four fish pathogenic bacteria were used in this study: *V. anguillarum* type strain LMG 4437 isolated from Atlantic cod (*Gadus morhua* L.) by J. Bagge [49]; *Vibrio alginolyticus* type strain V2 isolated from *Dentex dentex*, during outbreaks of vibriosis [50]; *V. harveyi* type strain VH2 isolated from farmed juvenile *Seriola dumerili* during outbreaks of vibriosis in Crete, Greece [51]; and *A. veronii* biovar sobria isolated from farmed European seabass in the Mediterranean Sea [21]. The isolates were kindly provided by Dr Pantelis Katharios from the Hellenic Center for Marine Research, Heraklion, Crete, Greece. All bacterial strains were cultured in 5 mL of tryptic soy broth (TSB) added 2% NaCl (w/v) at 25 °C. The culture period was 24 h for *V. anguillarum*, *V. alginolyticus*, and *A. veronii*, and 48 h for *V. harveyi*.

### 2.2. Antibacterial Assay by Use of Axenic Cultures

#### 2.2.1. Experiments at 25 PPT Salinity

Initially, bacterium-free microalgae cultures were obtained after treatment with a mixture of antibiotics (oxolinic acid 10 μg/mL, kanamycin 10 μg/mL, penicillin G 150 μg/mL, streptomycin 75 μg/mL) to kill bacteria present in the cultures [52]. This was verified by plating on tryptic soy agar (TSA) dishes with added 2% NaCl (*w*/*v*). The antibiotic was removed after successive dilutions over 3–4 weeks. The antimicrobial activity of *Tetraselmis* sp. (red var.), *Tetraselmis* sp. (red var., Pappas), *Tetraselmis* sp. (red var., Kotichi), *Tetraselmis* sp. (palmella), *T. marina* (var. Messolonghi), *A. gracilis* and *A. carterae* at a salinity of 25 ppt was studied. *Chlorella minutissima* was used as a reference species, as it has shown antibacterial properties in an earlier study [53]. Three fish pathogenic bacteria were used in this experiment: *V. anguillarum*, *A. veronii*, and *V. alginolyticus*. Sterile seawater at 25 ppt with added Walne’s growth medium was used as a control treatment.

The relationship between the concentration of pathogenic bacteria in terms of colony-forming units (CFU) per unit volume and OD_600_ was determined in a preliminary experiment. The population density (cells/mL) of microalgae in the axenic cultures was followed using a Neubauer-improved hemocytometer through a light microscope ZEISS Axio Imager.A2. After the algae cultures reached the late exponential phase, aliquots of 5 mL of each of the microalgae cultures were inoculated with different bacterial pathogens in test tubes at a final concentration of 10^4^ CFU/mL in four replicates for each combination of microalgae vs. pathogen. The growth of the added pathogens was followed 0, 1, 2, 4, and 6 days after inoculation by spreading ten-fold dilutions on TSA dishes, and the colonies were counted after incubation for 7 days [54]. The growth of the pathogens without the microalgae cultures was also tested (control). The experiments were performed both in the presence and absence of light.

#### 2.2.2. Experiments at Different Salinities

In a second series of experiments, *A. carterae*, *A. gracilis*, *Tetraselmis* sp. (red var.), *Tetraselmis* sp. (palmella), and *Tetraselmis* sp. (red var., Pappas) were selected because in the first series of experiments, these microalgae species were more efficient against the four pathogenic bacteria (*V. anguillarum*, *A. veronii*, *V. alginolyticus*, and *V. harveyi)*. In addition, microalgae *D. salina* was tested, as it had not been tested in the first series of experiments.

In this series of experiments, the axenic microalgae were cultured with aeration to obtain better growth, and each algae strain was tested at the salinity where it was originally isolated from the lagoons. So, the salinities used were 25, 100, 60, and 40 ppt for *A. carterae*, *A. gracilis*, *Tetraselmis* sp., and *D. salina*, respectively. At the start, the numbers of microalgae cells in the cultures were counted in a Neubauer-improved hemocytometer using a microscope as well as the concentration of pathogenic bacteria using a spectrophotometer. The numbers of cells in the microalgae cultures were measured also at the end of the experiment (day 6). 

The experimental procedure followed was the same as in the first experiment, where the growth of the pathogens in the microalgae cultures was monitored 0, 1, 2, 4, and 6 days after inoculation by spreading serial dilutions in plates with TSA and counting colonies 2–3 days after incubation [54]. The experiments were performed both in the presence and absence of light, in duplicate in each case with the microalgae *Chlorella minutissima* being used as the reference microalgae [52]. Pathogenic bacteria in 25 ppt sterile seawater added with Walne’s medium were used as a control.

### 2.3. Extracellular Antimicrobial Assay 

In these experiments, the eucaryotic microalgae *A. carterae*, *A. gracilis, Tetraselmis* sp. (red var. Pappas), and *Nephroselmis* sp. and the cyanobacteria *Phormidium* sp., *Anabaena* sp. and *Cyanothece* sp. were used. The fish pathogenic bacteria used were *V. anguillarum*, *A. veronii*, *V. alginolyticus*, and *V. harveyi*.

At the onset (day 0) of the experiment, the microalgae were counted under a microscope. The concentration of pathogenic bacteria was estimated using a spectrophotometer at 600 nm, and the bacteria were added to the experiment at a final concentration of 10^4^ CFU/mL. The presence of the pathogenic bacteria in the microalgae cultures was then verified by plating ten-fold dilutions of the microalgae cultures on TSA plates. 

The algae cells from an axenic culture at the exponential phase of the growing microalgae strains were separated from the culture medium by centrifugation (1 mL of microalgae in Eppendorf tubes), and there, the pellets were removed [55]. Briefly, the supernatants were obtained by centrifugation at 8000× *g* at 4 °C for 20 min using a SL8R centrifuge (Thermo Fisher Scientific, Osterode am Harz, Germany) and thereafter filter-sterilized (through 0.22 μm pore-size filters) [54].

The inhibitory activity was determined using 96-well ELISA plates. In each well, 150 μL of tryptic soy broth (TSB), 10 μL of each pathogen (diluted in sterile seawater 25 ppt added Walne’s medium), and 50 μL of the culture supernatant of each microalgae species were added. Autoclaved 25 ppt seawater was used as a negative control instead of culture supernatant. Each microalgae species was tested in four replicates where two samples were taken from two different cultures of 2 mL. The extracellular antimicrobial activity of cell-free supernatants was determined by measurement of optical density at 600 nm 0, 2, 4, 6, 21, 23, 25, 48, 72, 96, and 168 h after inoculation. The results were then modified using the following formula to calculate inhibition efficiency (*IE*):(1)IE=OD in presence of culture supernatantOD of negative control
where *IE* < 1 means that there was inhibition of pathogenic bacteria, *IE* = 1 means there was no inhibition, and *IE* > 1 means that the sample promoted the growth of pathogens [56].

### 2.4. Statistical Analysis

The statistical analysis included both an independent samples *t*-test and correlation analysis. For the *t*-test, we compared the mean values of numbers of CFU between the experimental microalgae cultures and controls. Prior to conducting the *t*-test, we checked the assumptions of normality and homogeneity variance by use of Kolmogorov–Smirnov and Levene’s test, respectively. Statistical significance was determined at a level of ≥95%, with *p* < 0.05 considered statistically significant. Additionally, a correlation analysis was conducted to examine the relationship between the presence of microalgae species (independent variable) and the concentration of pathogenic bacteria (dependent variable). The Pearson correlation coefficient (r) was applied to measure the strength and direction of the correlation. All statistical analyses were performed using IBM SPSS Statistics version 28.0.

## 3. Results 

### 3.1. Antibacterial Assay with Axenic Cultures

#### 3.1.1. Assay at 25 PPT Salinity

The numbers of pathogenic bacteria as determined by CFU counts on TSA dishes showed some instability during the first two days of the experiment but thereafter decreased for all tested pathogenic bacteria in all the microalgae cultures (Figure 1a–f). The microalgae that inhibited most efficiently the pathogens were *A. carterae*, *A. gracilis*, *Tetraselmis* sp. red var., *Tetraselmis* sp. (palmella), and *Tetraselmis* sp. (red var., Pappas). In the case of *Tetraselmis* sp. (red var., Pappas), it was observed that on the first two days of the experiment, it showed no antimicrobial activity against the pathogen *A. veronii* in either light or dark conditions. However, as the experiment progressed, the inhibition of the pathogen increased, reaching a greater than 75% reduction in the initial pathogen cell concentration. The percentages of reduction were generally higher in the presence of light, with an average of 8% higher effectiveness under light conditions (Figure 1e). Similarly, in the case of *Tetraselmis* sp. (red var., Pappas), the inhibition of *V. alginolyticus* was lower during the first two days of the experiment compared with the following days, and pathogen cell concentration showed no statistically significant difference from the control. In the case of *Tetraselmis* sp. (red var., Pappas) against *V. alginolyticus*, there was a strong inhibitory activity throughout the experiment, with more than a 93% reduction in the pathogenic cells in the presence of light (Figure 1f). There was no significant difference in the effectiveness of *Tetraselmis* sp. (red var., Pappas) in light versus dark (Figure 1c). Finally, *Tetraselmis* sp. (red var., Pappas) showed no antimicrobial activity against *V. anguillarum* during the first two days of the experiment in either light or dark conditions (Figure 1d). However, on subsequent days, the antimicrobial activity of *Tetraselmis* sp. (red var., Pappas) increased to more than 80% in both light and dark conditions with no significant difference between them. Overall, on the last experimental day, *Tetraselmis* sp. (red var., Pappas) showed no antimicrobial activity against the pathogen neither in the dark nor in the light (Figure 1a). 

*Tetraselmis* sp. (palmella) strain showed inhibitory activity against *A. veronii* on day 1, with similar percentages in light and dark conditions, around 77% (Figure 1e). In the following days, the reduction in *A. veronii* cells exceeded 90% of the initial concentration, with similar rates in both light and dark conditions. Against *V. anguillarum*, *Tetraselmis* sp. (palmella) was efficient from day 2 to day 4, with similar rates in both light and dark conditions (Figure 1e). Against *V. alginolyticus*, *Tetraselmis* sp. (palmella) was effective from the beginning of the experiment except for day 2 of the experiment (Figure 1f).

*Tetraselmis* sp. red var. cultures, in dark conditions, showed inhibitory activity against *V. alginolyticus* during the first two days of the experiment and again on day 6 of the experiment. In light conditions, the concentration of *V. alginolyticus* was significantly lower in the cultures of *Tetraselmis* sp. red var. compared with the control on days 2 and 6 of the experiment (*p* < 0.05) (Figure 1c). Against *V. anguillarum*, *Tetraselmis* sp. red var. had an inhibitory activity of 94% of the initial concentration in the dark and 97% in the light from day 1, and this activity increased as the experiment progressed (Figure 1a). Against *A. veronii*, the antimicrobial activity appeared on the second experimental day (day 1) and on day 4 (Figure 1b).

*A. gracilis* cultures showed inhibitory activity against *V. anguillarum* on day one after inoculation both in light and in dark conditions. In dark conditions, the *V. anguillarum* concentration was lower than the control on days 2 and 6 after inoculation (*p* < 0.05), while in light conditions, the *V. anguillarum* concentration was lower than on days 1, 2, and 6 (*p* < 0.05) (Figure 1a). Against *A. veronii*, both in light and dark conditions, *A. gracilis* showed inhibitory activity on experimental days 1 and 4 (Figure 1b). Against *V. alginolyticus*, the results were statistically significant on experimental days 1, 2, and 6. The results from *A. gracilis* are shown in the figures below (Figure 1c).

Finally, *A. carterae* cultures were effective against all pathogenic bacteria on day 2, with no significant difference in effectiveness between light and dark conditions (*p* < 0.05) (Figure 1a–c). 

#### 3.1.2. Experiments at Different Salinities under Aeration

In the second series of experiments, all microalgae tested—*A. carterae*, *A. gracilis*, *Tetraselmis* species (red var., palmella, red var. Pappas), and *D. salina*—reduced the growth of bacteria compared with the control treatments, in which the number of bacteria increased exponentially. On the 4th day of the experiment, the biggest differences were noted for all microalgae tested.

In the case of *V. anguillarum*, the mean density was 3.3 × 10^7^ CFU/mL, while in exposure to light, the microalgae *A. gracilis* and *Tetraselmis* species (red var., palmella, red var. Pappas) resulted in a concentration range of 2 × 10^4^–8.9 × 10^6^ CFU/mL. Similarly, in exposure to light, the microalgae *A. carterae* and *D. salina* resulted in a concentration range of 6.6 × 10^5^–1.7 × 10^6^ CFU/mL compared to the control treatment of 6.0 × 10^6^ CFU/mL. In the absence of light, the concentration of bacteria in *A. gracilis* and *Tetraselmis* species (red var., palmella, red var. Pappas) was 3.0 × 10^4^–7.7 × 10^6^ CFU/mL, while the range for the concentration of bacteria in *A. carterae* and *D. salina* was 7.8 × 10^5^–1.9 × 10^6^ CFU/mL. 

In the case of *A. veronii*, the control treatment showed a concentration of bacteria of 3.5 × 10^7^ CFU/mL. In exposure to light, the microalgae *A. gracilis* and *Tetraselmis* species (red var., palmella, red var. Pappas) resulted in a concentration range of 4.0 × 10^4^–3.7 × 10^6^ CFU/mL. Similarly, in exposure to light, the microalgae *A. carterae* and *D. salina* resulted in a concentration range of 2.5 × 10^6^–4.2 × 10^6^ CFU/mL compared to the control treatment of 8.1 × 10^6^ CFU/mL. In the absence of light, the concentration of bacteria in *A. gracilis* and *Tetraselmis* species (red var., palmella, red var. Pappas) was 1.6 × 10^5^–1.1 × 10^7^ CFU/mL, and the concentration range in *A. carterae* and *D. salina* was 2.6 × 10^6^–4.6 × 10^6^ CFU/mL. For *V. alginolyticus*, the control treatment showed a concentration of bacteria of 2.4 × 10^7^ CFU/mL. In exposure to light, the microalgae *A. gracilis* and *Tetraselmis* species (red var., palmella, red var. Pappas) resulted in a concentration range of 6.0 × 10^4^–4.0 × 10^5^ CFU/mL. Similarly, exposure to light and the microalgae *A. carterae* and *D. salina* resulted in a concentration range of 2.1 × 10^6^–2.8 × 10^6^ CFU/mL, while the control treatment had a concentration of 4.4 × 10^6^ CFU/mL. In the absence of light, the bacterial concentration range for *A. gracilis* and *Tetraselmis* species (red var., palmella, red var. Pappas) was 1.4 × 10^5^–1.1 × 10^7^ CFU/mL, and for *A. carterae* and *D. salina,* it was 3.0 × 10^6^–3.6 × 10^6^ CFU/mL.

In the case of *V. harveyi*, the control treatment exhibited a concentration of bacteria of 3.3 × 10^8^ CFU/mL, whereas in the presence of light, the range in *A. gracilis* and *Tetraselmis* species (red var., palmella, red var. Pappas) was between 1.0 × 10^4^ and 3.9 × 10^5^ CFU/mL. Similarly, the range of *A. carterae* and *D. salina* in the presence of light was between 2.3 × 10^5^ and 2.3 × 10^6^ CFU/mL, while the control treatment yielded 4 × 10^6^ CFU/mL. On the other hand, in the absence of light, the concentration of bacteria range for *A. gracilis*, and *Tetraselmis* species (red var., palmella, red var. Pappas) was between 2.1 × 10^5^ and 7.5 × 10^6^ CFU/mL, and the range for *A. carterae* and *D. salina* was between 2.1 × 10^6^ and 2.6 × 10^6^ CFU/mL.

During the entire experiment, *A. gracilis* exhibited the highest efficiency (Figure 2a,c), demonstrating a strong correlation, r = 0.902, and significantly reducing the concentration of *V. anguillarum* cells by over 94% (*p* < 0.05) of the initial concentration. This effect was observed in both light and dark conditions. *Tetraselmis* red var. Pappas exhibited a significant reduction in the concentration of *V. anguillarum* cells, particularly in light conditions starting from day 1, r = 0.8403). In dark conditions, except for day 2, *Tetraselmis* red var. Pappas demonstrated a notable decrease in cell concentration throughout the entire experiment (Figure 2a,c). *Tetraselmis* red var. and *Tetraselmis* palmella both reduced the concentration of *V. anguillarum* cells, with *Tetraselmis* red var. showing reductions from day 1 in both light and dark conditions and *Tetraselmis* palmella showing reductions from the beginning of the experiment in light and day 1 in darkness (Figure 2a,c). On days 1, 4, and 6, both in light and dark conditions, *D. salina* and *A. carterae* exhibited inhibitory activity, respectively (Figure 2b,d).

While the microalgae cultures showed inhibitory effects against *V. anguillarum*, they were less effective at reducing the concentration of *A. veronii* in our experiments. *A. gracilis* and *Tetraselmis* red var. Pappas showed the highest antimicrobial activity compared with the control. *A. gracilis* demonstrated a significant reduction in *A. veronii* cell concentration in both light and dark conditions. The reduction was observed starting on day 2 in light conditions and from the beginning of the experiment in darkness. Furthermore, an additional decrease in cell concentration was observed on day 2 (*p* < 0.05) (Figure 3a,c), with a correlation of r = 0.7043. Meanwhile, *Tetraselmis* red var. Pappas exhibited a significant reduction in *A. veronii* cell concentration, particularly in the light, starting on day 1 (*p* < 0.05) (Figure 3a,c). This reduction showed a correlation of r = 0.8161. *Tetraselmis* red var. showed inhibitory activity both in the light and in the dark on day 2 (Figure 3a,c). *Tetraselmis* palmella exhibited inhibitory activity both in the light and in the dark on day 2 (Figure 3a,c), while *D. salina* showed inhibitory activity on days 1, 4, and 6, both in the light and in the dark (Figure 3b,d). *A. carterae* showed inhibitory activity against *A. veronii* in both light and dark conditions, with a significant reduction in cell concentration observed, in light on day 2, and in the dark on days 1 and 2 of the experiment, respectively (Figure 3b,d). 

The results showed that *A. gracilis* showed a statistically significant difference in reducing the concentration of *V. alginolyticus* compared with the control. Specifically, in the light, a significant reduction in *V. alginolyticus* cell concentration was observed during the last two days of the experiment and on day 1 with a correlation of r = 0.863. Similarly, in the dark conditions, a significant reduction was observed during the last two days (*p* < 0.05) (Figure 4a,c). Regarding *Tetraselmis* red var., significant differences in reducing *V. alginolyticus* cell concentration were observed in the dark on days 1, 4, and 6, and in the light on days 4 and 6 of the experiment (*p* < 0.05) (Figure 4a,c). *Tetraselmis* palmella displayed inhibitory activity against *V. alginolyticus* in both light and darkness from day 2 (Figure 4a,c). *Tetraselmis* red var. Pappas exhibited significant inhibitory activity against *V. alginolyticus* with statistically significant differences observed in light conditions during the last two days of the experiment, showing a correlation of r = 0.7581. Additionally, in the dark conditions, significant inhibitory activity was observed on days 1, 4, and 6 (Figure 4a,c). *A. carterae* demonstrated statistically significant differences in inhibiting *V. alginolyticus* only on day 4 in light and day 6 in darkness, while *D. salina* showed significant differences in both light and dark conditions during the last two days of the experiment (Figure 4 a,c).

*A. gracilis*, *Tetraselmis* palmella, and *Tetraselmis* red var. were found to be the most effective cultures against *V. harveyi* throughout the experiment, exhibiting significant reductions in cell concentration in both light and dark conditions compared with the control. However, no clear inhibitory effects were observed on the first day (Figure 5a,c). *Tetraselmis* red var. Pappas showed inhibitory activity against *V. harveyi* with significant effects observed in the light conditions starting from day 1 and in the dark conditions starting from day 2 (Figure 5a,c). These results were supported by a correlation r = 0.5952. *A. gracilis* was effective also against *V. harveyi*, with a more than 94% reduction in the initial concentration in both light (r = 0.5171) and darkness throughout the duration of the experiment. *A. carterae* and *D. salina* had no inhibitory activity against *V. harveyi* (Figure 5b,d).

### 3.2. Extracellular Assay

Seven species of microalgae (*Tetraselmis* sp. (red var., Pappas), *Nephroselmis* sp., *A. gracilis*, *Phormidium* sp., *Anabaena* sp., *Cyanothece* sp., and *A. carterae*) were examined for extracellular antimicrobial activity using a spectrophotometer at 600 nm. 

Our results indicate that *Phormidium* sp. and *Anabaena* sp. inhibited the growth of the concentration of *V. anguillarum* cells only twenty-five hours after inoculation (*p* < 0.05), but they promoted growth between 48 and 96 h (*p* < 0.05) compared with the control. After 96 h of inoculation, growth was inhibited again, but the difference was not statistically significant (Figure 6a). *Cyanothece* sp. did not show inhibition of the growth of *V. anguillarum* cells. *A. carterae* demonstrated inhibitory activity between 48 and 72 h. In the experiment with *Tetraselmis* sp. (red var., Pappas), the growth of the pathogen was inhibited during the period from 6 to 21 h, with a strong correlation of r = 0.9904. *A. gracilis* exhibited an inhibition of *V. anguillarum* growth only two hours after inoculation. *Nephroselmis* sp. showed inhibition six hours after inoculation. Figure 6 shows the Inhibition Efficiency (*IE*) for *Tetraselmis* red var. Pappas, *Nephroselmis* sp., *A. gracilis*, *A. carterae*, *Phormidium* sp, *Anabaena* sp., and *Cyanothece* sp., against *V. anguillarum.* The *IE* values were calculated using the modified photometric measurements of Equation (1). 

In general, our results indicated that *A. veronii* was more resistant to all the micro-algae strains tested compared with the control other pathogenic bacteria. However, its resistance to inhibition decreased over time. Nonetheless, this reduction in resistance is not notably significant, since there is also a decrease in absorption values in the control samples. The low absorption values at the end of the experiment suggest that the activity of *A. veronii* gradually weakened regardless of whether an external agent was present. Figure 6b illustrates the inhibition efficiency (IE) for the cultures at *OD* 600 nm. In general, cyanobacteria did not demonstrate inhibition efficiency against *A. veronii*. At the onset of the experiment, only *A. gracilis* and *Nephroselmis* sp. demonstrated statistically significant inhibition efficiency (*p* < 0.05). Among them, *A. gracilis* exhibited a correlation of r = 0.829.

Our experiments with *V. alginolyticus* yielded promising results compared with the control treatment, particularly for the cyanobacteria strains. The inhibitory effect on *Phormidium* sp. and *Anabaena* sp. was observed during the period from 21 to 48 h, and for *Cyanothece* sp., it was observed during the period from 25 to 48 h (*p* < 0.05). The samples containing *Tetraselmis* sp. (red var., Pappas) with *V. alginolyticus* showed statistically significant inhibition between 4 and 25 h with a strong correlation of r = 0.9946. These results indicate that all the tested cyanobacteria and *Tetraselmis* sp. (red var., Pappas) produced compounds that accumulated and inhibited the activity of *V. alginolyticus* over time with inhibition activity peaking at 21 h for *Tetraselmis* sp. (red var., Pappas). *A. gracilis* showed inhibition at 72 h, while *Nephroselmis* sp. had a peak inhibition at 48 h. However, *A. carterae* did not show statistically significant inhibition efficiency against *V. alginolyticus*. The results of inhibition efficiency (*IE*) against *V. alginolyticus* are illustrated in Figure 6c.

The results of our experiments with *V. harveyi* revealed a statistically significant inhibition of growth for the *Phormidium* sp. strain at 96 h (*p* < 0.05) compared with the control treatment. *Anabaena* sp. showed inhibition between 23 and 25 h. However, no inhibition was observed for *Cyanothece* sp. In our experiments with the *Tetraselmis* sp. (red var., Pappas) strain, inhibition against *V. harveyi* was observed between 21 and 23 h with a strong correlation of r = 0.981. *A. carterae* demonstrated inhibition at 48 h, and *A. gracilis* demonstrated inhibition at 72 h. *Nephroselmis* sp. exhibited inhibition against *V. harveyi* between 4 and 6 h. The inhibition efficiency (*IE*) of the samples against *V. harveyi* is illustrated in Figure 6d. 

## 4. Discussion

In this study, experiments were conducted to assess the antimicrobial activity of twelve microalgae species isolated from lagoons in western Greece against four fish pathogenic bacteria. The experiments were carried out at first at 25 ppt salinity and under ideal axenic culture conditions with aeration, and then, an extracellular assay was performed. 

The results from the first series of experiments showed that all microalgae cultures studied at 25 ppt salinity exhibited inhibitory activity against the three tested pathogens (*V. anguillarum*, *A. veronii*, *V. alginolyticus*) with varying degrees of efficiency. Regarding the effect of light, we observed that the inhibitory activity of most microalgae cultures was generally higher in the presence of light. The average efficiency was approximately 8% higher under light conditions. This finding suggests that light may play a role in enhancing the antimicrobial activity of microalgae cultures. In terms of the vulnerability of the tested pathogens, our results showed that *A. carterae*, *A. gracilis*, *Tetraselmis* sp. red var., *Tetraselmis* sp. (palmella), and *Tetraselmis* sp. (red var., Pappas) were the most efficient at inhibiting the pathogens. Among these, *Tetraselmis* sp. (red var., Pappas) and *Tetraselmis* sp. (palmella) were effective against all three tested pathogens. However, we also observed some variation in the inhibitory activity of the microalgae cultures against different pathogens. 

The second series of experiments at different salinity conditions showed that all microalgae tested, including *A. carterae*, *A. gracilis*, *Tetraselmis* species (red var., palmella, red var. Pappas), and *D. salina*, reduced the growth of bacteria (*V. anguillarum*, *A. veronii*, *V. alginolyticus*, *V. harveyi*) compared with the control treatments, in which the number of bacteria increased exponentially. The microalgae were particularly effective in reducing the concentration of *V. anguillarum* cells with *A. gracilis* proving to be the most effective treatment, reducing the concentration of *V. anguillarum* cells by over 94% in both light and dark conditions with a strong correlation of r = 0.902. *Tetraselmis* red var. Pappas also significantly reduced the concentration of *V. anguillarum* cells particularly in light conditions from day 1 (r = 0.8403). Additionally, it exhibited a significant reduction in dark conditions from the beginning of the experiment except for day 2. *Tetraselmis* red var. and *Tetraselmis* palmella also reduced the concentration of *V. anguillarum* cells. On days 1, 4, and 6, both in light and dark conditions, *D. salina* and *A. carterae* exhibited inhibitory activity. Overall, the microalgae cultures showed inhibitory effects against bacteria, particularly *V. anguillarum*. *A. gracilis* was effective also against *V. harveyi* with more than a 94% reduction in the initial concentration in both light (r = 0.5171) and darkness throughout the duration of the experiment. However, its effectiveness against *A. veronii* was lower with inhibitory activity beginning in light conditions on day 2 and essentially in darkness on day 2. *Tetraselmis* sp. red var. was also effective against *V. alginolyticus*, with a more than 99% reduction in the initial concentration compared with the control in both light and dark conditions, and *V. harveyi*, but it was less effective against *A. veronii*. Against *A. veronii,* it displayed inhibitory activity both in light and dark conditions on day 2. *Tetraselmis* sp. (palmella) showed strong inhibitory activity against all four pathogenic bacteria with the highest effectiveness against *V. harveyi*. *A. carterae* had lower effectiveness compared with the other microalgae species, but it still showed inhibitory activity against *V. anguillarum* and *A. veronii*, with activity beginning on day 1. Finally, *D. salina* showed only slight inhibitory activity against *V. anguillarum*, *A. veronii*, and *V. alginolyticus*. 

The differences between the first two experiments that were performed with microalgal cultures were firstly the experimental setup, where aeration was added only in the second experiment. Furthermore, in the first experiment, all cultures were grown in 25 ppt salinity conditions, while in the second experiment, different salinity conditions were used depending on the type of microalgae. Another difference was in the species of microalgae used. In the first experiment, *Tetraselmis* sp. (red var.), *Tetraselmis* sp. (red var., Pappas), *Tetraselmis* sp. (red var., Kotichi), *Tetraselmis* sp. (palmella), *T. marina* (var. Messolonghi), *A. gracilis,* and *A. carterae* were used. In the second experiment, *Tetraselmis* sp. (red var.), *Tetraselmis* sp. (red var. Pappas), *Tetraselmis* sp. (palmella), *Dunaliella salina, A. gracilis,* and *A. carterae* were utilized. Lastly, the type of bacteria used also differed between the two experiments. In the first experiment, *V. anguillarum, A. veronii,* and *Vibrio alginolyticus* were added, while in the second experiment, *Vibrio harveyi* was also included. 

In the first series of experiments, all microalgae cultures studied exhibited inhibitory activity against the three tested pathogens, *V. anguillarum*, *A. veronii*, and *V. alginolyticus*, with varying degrees of effectiveness. Interestingly, it was observed that the inhibitory activity of most microalgae cultures was generally higher in the presence of light, which suggests that light may play a role in enhancing the antimicrobial activity of microalgae cultures. Free oxygen radicals were probably produced during the process of photosynthesis, and this may have increased the vulnerability of pathogenic bacteria. Additionally, *A. carterae*, *A. gracilis*, *Tetraselmis* sp. red var., *Tetraselmis* sp. (palmella), and *Tetraselmis* sp. (red var., Pappas) were the most efficient in inhibiting the pathogens, with *Tetraselmis* sp. (red var., Pappas) and *Tetraselmis* sp. (palmella) being effective against all three tested pathogens. In the second series of experiments conducted at different salinity conditions, all microalgae tested, including *A. carterae*, *A. gracilis*, *Tetraselmis* species (red var., palmella, red var. Pappas), and *D. salina*, reduced the growth of bacteria compared with the control treatments, in which the number of bacteria increased exponentially. All microalgae tested were particularly effective at reducing the concentration of *V. anguillarum* cells in both light and dark conditions. In this second experiment, *V. harveyi* was the most resistant pathogen. 

The results of the extracellular assay indicated that several microalgae species showed inhibition of the growth of the pathogenic bacteria (*V. anguillarum*, *A. veronii*, *V. alginolyticus*, *V. harveyi*), while others promoted the growth or do not show inhibition compared with the control. The species that exhibited the highest inhibition efficiency against these bacteria varied depending on the specific pathogen being tested. Regarding *V. anguillarum*, *Phormidium* sp. and *Anabaena* sp. inhibited the growth of the pathogenic bacteria 25 h after inoculation, while *Cyanothece* sp. did not show any inhibitory activity. *A. carterae* exhibited inhibitory activity between 48 and 72 h after inoculation, while *A. gracilis* and *Nephroselmis* sp. showed inhibition 2 and 6 h after inoculation, respectively. *Tetraselmis* sp. (red var., Pappas) was found to inhibit the growth of *V. anguillarum* during the period from 6 to 21 h after inoculation (r = 0.9904). Concerning *A. veronii*, our results suggest that it was more resistant to all microalgae strains tested compared to the control. However, its resistance to inhibition decreased over time, suggesting that its activity gradually weakened regardless of whether an external agent was present. Only *A. gracilis* and *Nephroselmis* sp. exhibited statistically significant inhibition efficiency against *A. veronii* at the onset of the experiment. Our experiments with *V. alginolyticus* yielded promising results compared with the control treatment particularly for the cyanobacteria strains. *Phormidium* sp. and *Anabaena* sp. exhibited inhibitory activity during the period from 21 to 48 h, while *Cyanothece* sp. showed inhibitory activity between 25 and 48 h. The samples containing *Tetraselmis* sp. (red var., Pappas) with *V. alginolyticus* showed statistically significant inhibition between 4 and 25 h r = 0.9946). *A. gracilis* exhibited inhibition 72 h after inoculation, while *Nephroselmis* sp. had a peak inhibition 48 h after inoculation. However, *A. carterae* did not show statistically significant inhibition efficiency against *V. alginolyticus*. In the case of *V. harveyi*, *Phormidium* sp. exhibited a statistically significant inhibition of growth at 4 days compared with the control treatment. *Anabaena* sp. showed inhibition between 23 and 25 h, while *Tetraselmis* sp. (red var., Pappas) showed inhibition between 21 and 23 h (r = 0.981). *A. carterae* exhibited inhibition 48 h after inoculation, while *A. gracilis* exhibited inhibition 72 h after inoculation. *Nephroselmis* sp. exhibited inhibition against *V. harveyi* between 4 and 6 h. However, no inhibition was observed for *Cyanothece* sp. 

The extracellular assay experiment and the previous two experiments are different in terms of their methods and results. The extracellular assay measures the effect of microalgae on the growth of pathogenic bacteria outside the cells, while the previous experiments focused on the effect of live microalgal cells on bacterial enzyme activity (proteases) or biofilm formation. In the extracellular assay, some microalgae species were found to inhibit the growth of pathogenic bacteria, while others promoted growth or had no effect. For example, all microalgae tested with the extracellular assay were effective against *V. alginolyticus*, while *A. veronii* was more resistant to all microalgae strains tested. In the previous two experiments, *V. anguillarum* was the most vulnerable pathogen for all microalgae tested, while *V. harveyi* was the most resistant.

The mechanisms of antimicrobial activity are not fully understood. However, we hypothesize that the inhibition of pathogens at the first experiments at 25 ppt and under ideal salinities was dependent upon the physical presence of the microalgal cells themselves. For example, the microalgal cells may compete with the bacteria for nutrients or resources, or they may physically block the attachment of bacterial cells to surfaces. On the other hand, the inhibition of pathogens at the extracellular assay may be due to secondary metabolites produced by microalgae during their metabolism. The accumulation of these secondary metabolites may be responsible for the observed antibacterial activity, as these compounds can be released into the extracellular environment and could potentially inhibit the growth or attachment of pathogenic bacteria. 

Microalgae excrete in their microenvironment a vast spectrum of metabolites that may influence microorganisms. The production of such compounds depends upon the culture conditions [57,58]. Omics, including transcriptomics, proteomics, and metabolomics represent an approach which may lead to the discovery of bioactive compounds produced by microalgae, which could not be detected for the past decades due to limited coverage and resolution of the conventional methods. Transcriptomics focuses on the expression pattern of genomes, proteomics on the protein profile, and the metobolomics on the metabolic pathways that dominate under different culture conditions. Microalgae produced compounds with antimicrobial activity such as peptides, alkaloids, flavonoids, and fatty acids [59]. 

Previous research supports the findings of this study, demonstrating the presence of bioactive substances in many of the examined microalgae and suggesting their use in fish hatcheries. *Tetraselmis* sp. (red var.) and *A. gracilis*, for instance, have been noted for their beneficial properties [60,61,62,63,64,65,66]. The genus *Tetraselmis* produces a variety of carotenoids, such as β-carotene, lutein, and biolaxanthin [63], while *A. gracilis* has been suggested for use in fish hatcheries due to the stability of their cultures and the ability to eliminate contaminants through increased salinity [66]. *Nephroselmis* sp. is a natural source of antioxidants due to its high content of carotenoids (neoxanthin, lycopene, xanthophylls, lutein, β-carotene) and siphonaxanthin [67,68], an unusual dye with potential biotechnological applications [69], as well as lipids. *A. carterae* produces amphidinols, carotoxins, and fatty acids (EPA and DHA) with nutritional and pharmaceutical applications [70,71]. It has been used as a reference microalgal species for numerous genetic and physiological studies [71,72] and can be mass-cultivated in a photobioreactor [70]. *Phormidium* sp. produces useful components such as antioxidant carotenoids and phycovilins (large amounts of phycocyanin), which have pharmaceutical uses [70]. *Anabaena* sp. contains three main biliproteins, two of which (C-phycocyanin and allophycocyanin) are found in all cyanobacteria, while the third (phycoerythrocyanine, λmax~ 568 nm) does not occur in other cyanobacteria [73]. Further studies are needed to identify the specific compounds responsible for the inhibitory activity.

In this study, we found that antimicrobial activity was consistent regardless of whether the experiments were conducted in light or dark conditions. This suggests that antimicrobial activity is related to substances within the microalgae and warrants further investigation to identify these specific substances. Regarding the mechanisms of antimicrobial action in extracellular assay, it is possible that the microalgae species are producing extracellular compounds that inhibit the growth of the pathogenic bacteria. Further research would be necessary to identify the specific compounds responsible for the observed inhibitory activity. Our results also contradict the possibility that the antimicrobial action in the extracellular assay was due to reactive oxygen species (ROS) formed during photosynthesis or oxygen breakdown in the cultures [74,75]. While ROS do exhibit antimicrobial activity by attacking a range of targets in various pathogenic microorganisms, our research suggests that this is not the case for the microalgae studied. This conclusion is supported by the fact that our samples were kept in the dark, which suggests that the observed antibacterial activity was not dependent on light-induced ROS generation. 

Overall, our results suggest that microalgae from lagoons in western Greece could potentially be a source of natural antimicrobial compounds with applications in aquaculture and other industries. The inhibitory activity varies depending on the microalgae species and the tested pathogens. Light appears to enhance the antimicrobial activity of microalgae cultures. Further research would be necessary to identify the specific compounds responsible for the observed inhibitory activity and to assess their safety and effectiveness in real-world applications.

## 5. Conclusions

In the axenic culture experiments, *Tetraselmis* sp. red var. *Tetraselmis* sp. (red var., Pappas) and *A. gracilis* showed antimicrobial activity against all four tested pathogens (*V. anguillarum*, *A. veronii*, *V. alginolyticus*, and *V. harveyi*). Based on the results of the extracellular antimicrobial assay, it appears that *Tetraselmis* sp. (red var., Pappas) was the most effective against *V. anguillarum* and *V. alginolyticus*, the three cyanobacteria were effective against *V. alginolyticus*, and *Tetraselmis* sp. (red var., Pappas), *A. gracilis*, *Phormidium* sp., and *Anabaena* sp., were effective against *V. harveyi*. 

This study presented data on the antimicrobial properties of specific microalgae species isolated from lagoons in western Greece, which could be used in aquaculture. The mass production of microalgae in Greece presents an opportunity for new companies to exploit local species to produce value-added products for aquaculture, such as antioxidants, food additives, pharmaceuticals, and even biofuels [76,77]. 

One limitation of our study is that we only tested the antimicrobial activity of these microalgae species against three specific pathogenic bacteria. It would be useful to conduct additional studies to assess the antimicrobial activity of these microalgae species against a wider range of pathogenic bacteria as well as determine the mechanisms by which they exert their antimicrobial activity. Additionally, the study did not evaluate the potential toxicity of the microalgae species to fish, which would be important information to consider in the context of aquaculture. Further research is needed to confirm these findings and to determine the optimal conditions for using these microalgae species in aquaculture.

Aquaculture is faced with numerous threats from diseases such as vibriosis [78], making it essential to find environmentally friendly alternatives to prevent fish pathogens [56]. These findings suggest that these microalgae species have the potential to be used as an alternative to antibiotics in aquaculture, which could help to reduce the risk of antibiotic resistance and the negative impacts of antibiotics on fish and human health as well as on the environment.

## Figures and Tables

**Figure 1 microorganisms-11-01396-f001:**
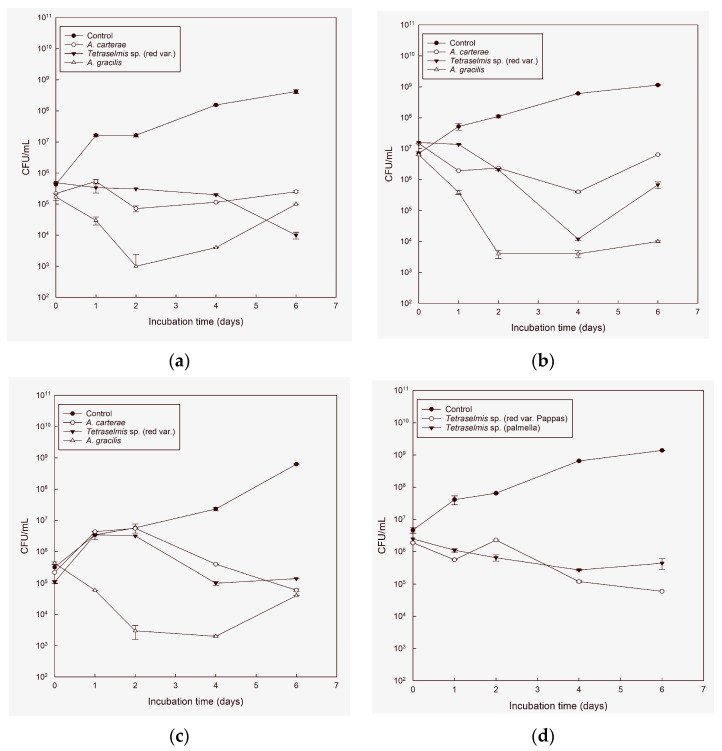
The graph shows the colony-forming units (CFU) per mL, average ± SE (*n* = 4) (log scale) of *V. anguillarum* (**a**,**d**), *A. veronii* (**b**,**e**), *V. alginolyticus* (**c**,**f**) in cultures of *Tetraselmis* sp. (red var.), *Tetraselmis* sp. (red var., Pappas), *Tetraselmis* sp. (red var., Kotichi), *Tetraselmis* sp. (palmella), *T. marina* (var. Messolonghi), *A. gracilis* and *A. carterae* compared with sterile seawater 25 ppt added Walne’s medium (control), over time in light conditions. The microalgae that inhibited the pathogens most efficiently were *A. carterae*, *A. gracilis*, *Tetraselmis* sp. red var., *Tetraselmis* sp. (palmella), and *Tetraselmis* sp. (red var., Pappas).

**Figure 2 microorganisms-11-01396-f002:**
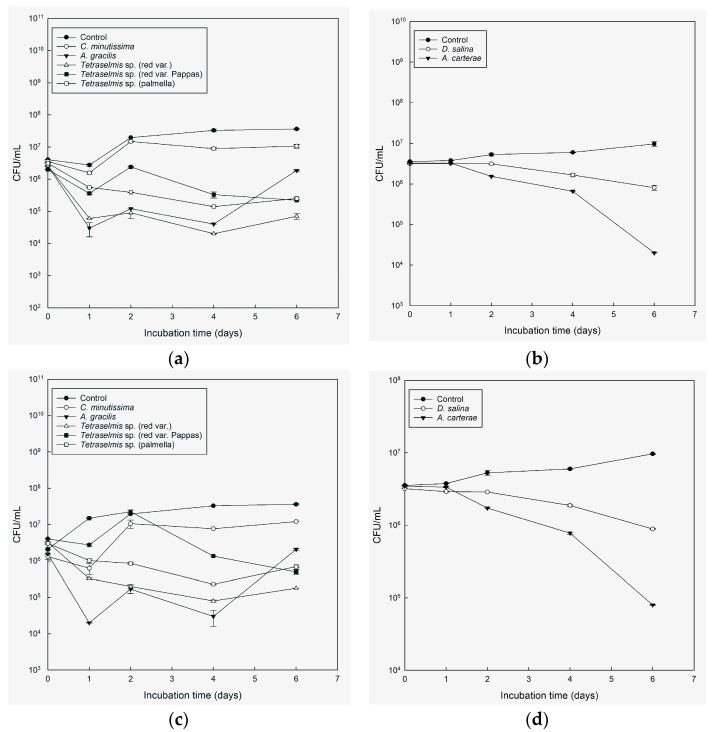
Colony-forming units (CFU) per mL, average ± SE (*n* = 2) (log scale) of *V. anguillarum* in cultures of *Chlorella minutissima*, *A. carterae*, *A. gracilis*, *Tetraselmis* species (red var., palmella, red var. Pappas), *D. salina*, compared with sterile seawater 25 ppt added Walne’s medium (control), through time in light conditions (**a**,**b**) and in the absence of light (**c**,**d**).

**Figure 3 microorganisms-11-01396-f003:**
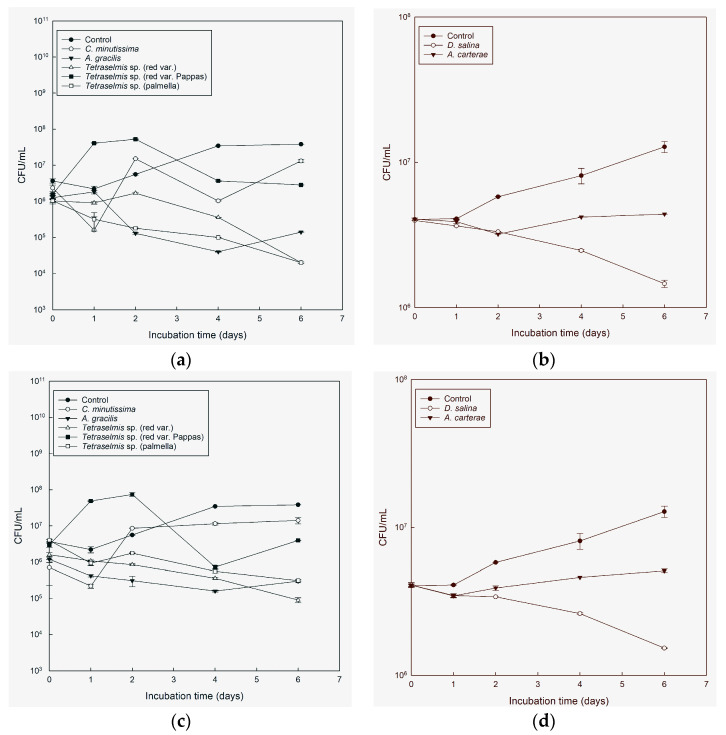
Colony-forming units (CFU) per mL, average ± SE (*n* = 2) (log scale) of *A. veronii* in cultures of *C. minutissima*, *A. carterae*, *A. gracilis*, *Tetraselmis* species (red var., palmella, red var. Pappas), *D. salina*, compared with sterile seawater 25 ppt added Walne’s medium (control), through time in light conditions (**a**,**b**) and in the absence of light (**c**,**d**).

**Figure 4 microorganisms-11-01396-f004:**
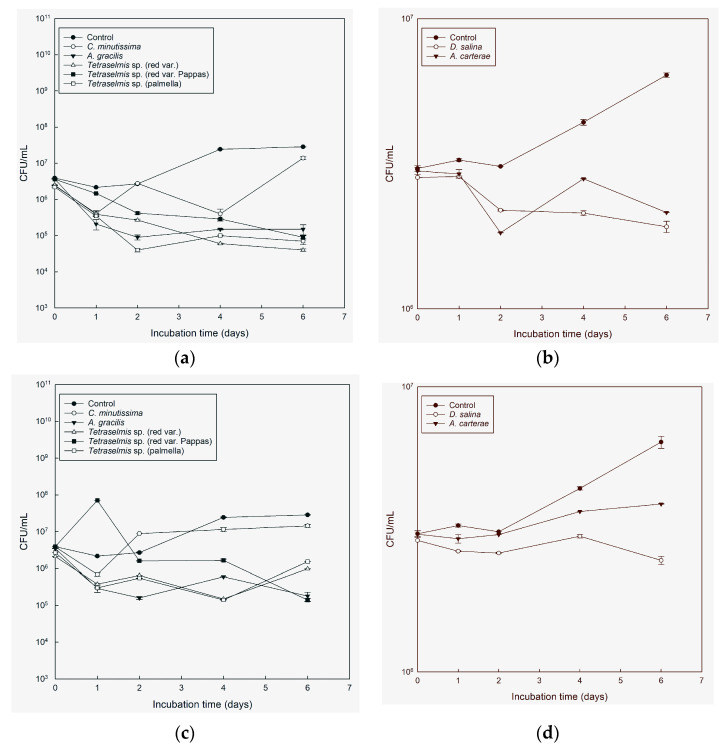
Colony-forming units (CFU) per mL, average ± SE (*n* = 2) (log scale) of *V. alginolyticus* in cultures of *C. minutissima*, *A. carterae*, *A. gracilis*, *Tetraselmis* species (red var., palmella, red var. Pappas), *D. salina*, compared with sterile seawater 25 ppt added Walne’s medium (control), through time in light conditions (**a**,**b**) and in the absence of light (**c**,**d**).

**Figure 5 microorganisms-11-01396-f005:**
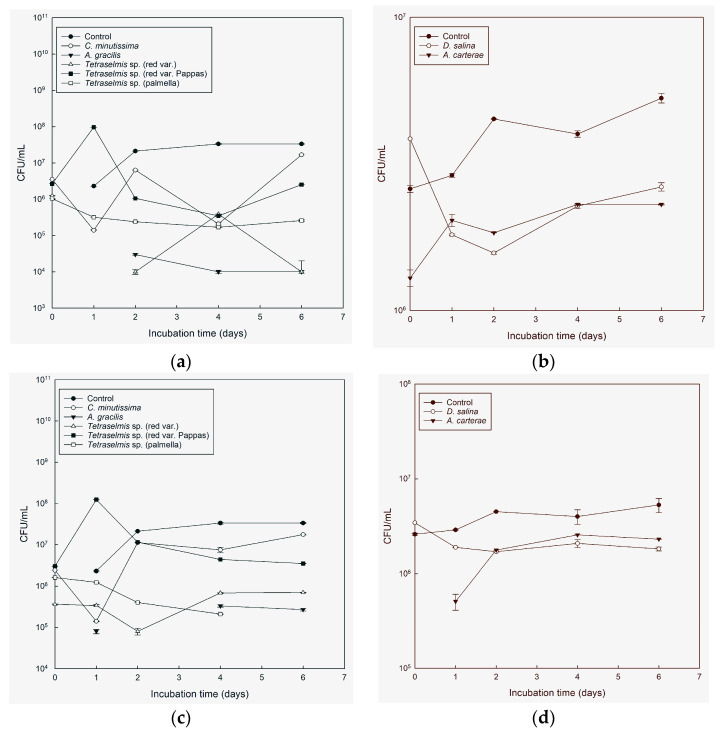
Colony-forming units (CFU) per mL, average ± SE (*n* = 2) (log scale) of *V. harveyi* in cultures of *C. minutissima*, *A. carterae*, *A. gracilis*, *Tetraselmis* species (red var., palmella, red var. Pappas), *D. salina*, compared with sterile seawater 25 ppt added Walne’s medium (control), through time in light conditions (**a**,**b**) and in the absence of light (**c**,**d**).

**Figure 6 microorganisms-11-01396-f006:**
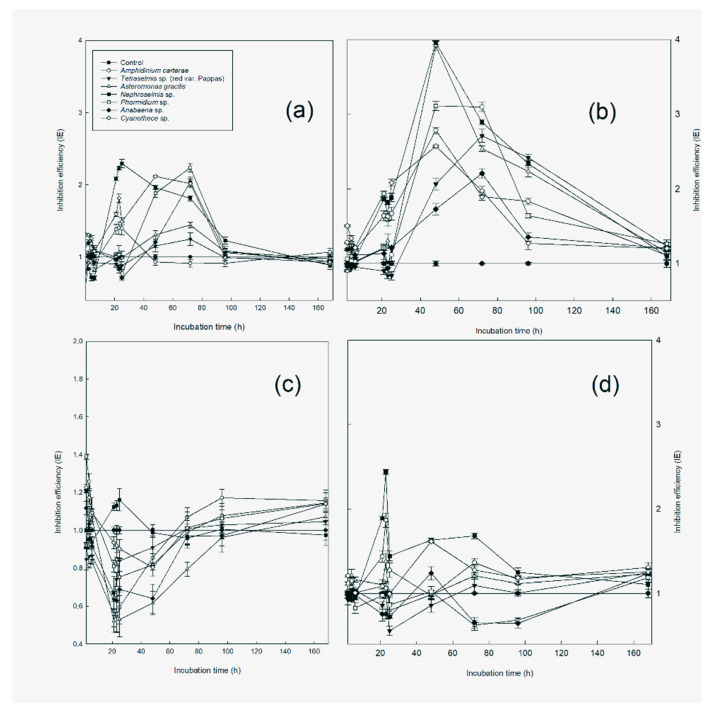
Inhibition efficiency of *Tetraselmis* sp. (red var., Pappas), *Nephroselmis* sp., *A. gracilis*, *Phormidium* sp., *Anabaena* sp., *Cyanothece* sp., *A. carterae*, against *V. anguillarum* (**a**), *A. veronii* (**b**), *V. alginolyticus* (**c**), *V. harveyi* (**d**) through time at 600 nm (*OD* 600 nm). Average ± SE (*n* = 4).

## Data Availability

The data presented in this study are available on request from the corresponding author.

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
