# Peer review of "Antibacterial Activity against Four Fish Pathogenic Bacteria of Twelve Microalgae Species Isolated from Lagoons in Western Greece"

_microorganisms, 2023, doi:10.3390/microorganisms11061396_

Round 1
Reviewer 1 Report
Reviewer’s comments
Title: Antibacterial activity in twelve microalgae species isolated from lagoons in Western Greece
Journal: Microorganisms
Manuscript Number: 2397485
Type of the Paper: Article
The research work entitled “Antibacterial activity in twelve microalgae species isolated from lagoons in Western Greece” presents the utilization and comparison of 12 kinds of microalgae isolated from natural environment against 4 common pathogenic bacteria in aquaculture. Compared with Compared with other previous anti-bacterial infection strategies, this study provides a more sustainable strategy to face the problems caused by the abuse of antibiotics. However, some concepts and information presented in this article are not fully expressed. I still believe that this work can make a great contribution for solving the problem of antibiotic abuse in aquaculture. I suggested this work may be “major revision” for publication in “Microorganisms”. Specific comments and general comments are given below:
Specific comments
1. Biological control is an ecologically friendly approach for the cultural system to counteract the bacterial infection. I think this study is a good example of biological control, therefore, I suggest that the author mention the relevant background in the Introduction section.
2. Authors are advised to change all scientific notation within the manuscript to a standard format. For example, 33×106 CFU/mL à 3.3×107 CFU/mL; 16,6×105 CFU/mL à 1.66×106 CFU/mL.
3. In all Figures, I believe that the author obtained the results through a rigorous experimental design and biological triplicate, but there is no error bar in all the figures. Authors are required to indicate error bars on all figures.
4. In order to increase readability, I hope the author can consider integrating Fig.6-Fig.9 into one figure. This will make it easier to compare the antimicrobial properties and relationships of the 12 microalgae.
5. I suggest that the author conduct additional Correlation Analysis on the collected data to show more evidence of the antibacterial advantages of Asteromonas gracilis and Tetraselmis sp. (red var., Pappas). This makes it easier to convince readers and achieve effective promotional effects.
6. In the discussion, I suggest that the authors use previous omics analysis (e.g., transcriptome, proteome, and metabolome) in other green algae (or other photosynthetic organisms) under light/dark cultures to illustrate some possible antimicrobial secretions or related biochemical pathway.
General comments
1. Line 49, The use of antibiotics in aquaculture à The abuse of antibiotics in aquaculture; a suggestion for a more precise word.
2. Line 69, …such as Vibrio alginolyticus, and Vibrio harveyi, cause… à …such as Vibrio alginolyticus and Vibrio harveyi, cause… (remove comma)
3. Line 144-145, … 24 h for Vibrio anguillarum, Vibrio alginolyticus and Aeromonas veronii, and 48 h for Vibrio harveyi. à … 24 h for Vibrio anguillarum, Vibrio alginolyticus, and Aeromonas veronii, and 48 h for Vibrio harveyi. (comma missing)
4. In Fig. 2, 3, 4, and 5, All labels of Tetraselmis sp. (palmella) have a slightly smaller font size than other species names, please present them consistently.
Author Response
Specific comments
- Biological control is an ecologically friendly approach for the cultural system to counteract the bacterial infection. I think this study is a good example of biological control, therefore, I suggest that the author mention the relevant background in the Introductionsection.
ANSWER. We added a section in the introduction where we mention biological control and the approaches mainly applied in aquaculture (Lines 117-119)
- Authors are advised to change all scientific notationwithin the manuscript to a standard format. For example, 33×106 CFU/mL à 3.3×107 CFU/mL; 16,6×105 CFU/mL à 1.66×106 CFU/mL.
ANSWER. Thank you for comment, we changed all scientific notation to standard format as suggested by the reviewer throughout the paper.
- In all Figures, I believe that the author obtained the results through a rigorous experimental design and biological triplicate, but there is no error bar in all the figures. Authors are required to indicate error bars on all figures.
ANSWER. Thank you for your comment. We included the error bars in all figures.
- In order to increase readability, I hope the author can consider integrating Fig.6-Fig.9 into one figure. This will make it easier to compare the antimicrobial properties and relationships of the 12 microalgae.
ANSWER. You are right, we integrated all in one figure (Figure 6) as you suggested.
- I suggest that the author conduct additional Correlation Analysis on the collected data to show more evidence of the antibacterial advantages of Asteromonas gracilisand Tetraselmis sp. (red var., Pappas). This makes it easier to convince readers and achieve effective promotional effects.
ANSWER. We appreciate your comment. We conducted the additional correlation analysis computing Pearson’s correlation coefficient (r) to measure the strength and direction of the correlation as mentioned in the methods section (Lines 231-241) and shown in the results.
- In the discussion, I suggest that the authors use previous omics analysis (e.g., transcriptome, proteome, and metabolome) in other green algae (or other photosynthetic organisms) under light/dark cultures to illustrate some possible antimicrobial secretions or related biochemical pathway.
ANSWER. We have included in the discussion following the suggestion of the reviewer (Lines 621-630)
General comments
- Line 49, The use of antibiotics in aquaculture à The abuse of antibiotics in aquaculture; a suggestion for a more precise word.
ANSWER. Thank you for your suggestion, we changed it to: The misuse of antibiotics in aquaculture… (Line 49)
- Line 69, …such asVibrio alginolyticus, and Vibrio harveyi, cause… à …such as Vibrio alginolyticus and Vibrio harveyi, cause… (remove comma)
ANSWER. We have removed the comma.
- Line 144-145, … 24 h for Vibrio anguillarum, Vibrio alginolyticus and Aeromonas veronii, and 48 h for Vibrio harveyi. à … 24 h for Vibrio anguillarum, Vibrio alginolyticus, and Aeromonas veronii, and 48 h for Vibrio harveyi. (comma missing)
Answer. We have inserted the comma.
- In Fig. 2, 3, 4, and 5, All labels of Tetraselmissp. (palmella) have a slightly smaller font size than other species names, please present them consistently.
ANSWER. We have made the necessary corrections.
Reviewer 2 Report
The MS use two methods to evaluate the antibacterial activity of 12 microalgae against selected four fish pathogenic bacteria. It obtained some meaning information. However, there are still some points need to modify.
1.Title, ‘of’ instead of ‘in’. also, suggest to “Antibacterial activity against four fish pathogenic bacteria of twelve microalgae species isolated from lagoons in Western Greece”
2.Too many keywords not closely related. Please consider to delete some irrelevant ones.
3.Introduction, too many paragraphs (seven). Please consider to combine some paragraphs.
4.Abstract, please add the detailed demonstration for the results obtained in this study. The current description for the results are too general to be understand for the readers.
5.Materials and methods, the subtitle of 2.1 is missing, and the structure of this section need to be re-arranged to make it clear to read, and the format of the subtitle of 2.2 is inconsistent with the others. Please check.
6.Consider to list the statistical analysis as a separate subsection in Materials and methods.
7.Figure 1, 2. Some demonstration for the results should be also added in the Figure legend, not only in the text.
8.Line 148-151, the authors obtained bacterium-free microalgae cultures after antibiotics treatment, and verified by plating on TSA dishes. I don’t think this method can obtain an absolute bacterium-free microalgae culture. An additional bacterial primer-based (such as 16S rRNA gene) PCR method is suggested to verity the bacterium-free status of the tested microalgaes.
9.The English in the MS need to improve by a native English speaker.
10.The starting time in Figure 8 is somewhat chaotic. It is recommended to change the legend to make the image clearer and cleaner.
11.Line 311, the expression of microbial concentration is wrong. Please check and correct similar problems.
12.IE is for inhibition efficacy or inhibition efficiency? Please check.
13.It is suggested to illustrate the results of significance test in the legend.
14.If possible, extract the bioactive compounds form the algal culture, and make sure the chemical structures and antibacterial activity quantitatively.
15.The format of the some reference is incorrect, for example Ref 12. please recheck.
The English in the MS need to improve by a native English speaker.
Author Response
1.Title, ‘of’ instead of ‘in’. also, suggest to “Antibacterial activity against four fish pathogenic bacteria of twelve microalgae species isolated from lagoons in Western Greece”.
ANSWER. We changed the title to “Antibacterial activity against four fish pathogenic bacteria of twelve microalgae species isolated from lagoons in Western Greece”.
2.Too many keywords not closely related. Please consider to delete some irrelevant ones.
AMSWER. We changed them to the keywords: antibacterial activity; fish pathogens; aquaculture; biological control
3.Introduction, too many paragraphs (seven). Please consider to combine some paragraphs.
ANSWER. Thank you for your comment. We combined them to five paragraphs.
4.Abstract, please add the detailed demonstration for the results obtained in this study. The current description for the results are too general to be understand for the readers.
ANSWER. We included more specific results.
5.Materials and methods, the subtitle of 2.1 is missing, and the structure of this section need to be re-arranged to make it clear to read, and the format of the subtitle of 2.2 is inconsistent with the others. Please check.
ANSWER. Thank you for bringing this to our attention. We changed it to:
2.1 Microorganisms and growth conditions (Line 131)
2.1.1 Microalgae cultures (Line 132)
6.Consider to list the statistical analysis as a separate subsection in Materials and methods.
ANSWER. We included a paragraph for statistical analysis in Lines 231-241.
7.Figure 1, 2. Some demonstration for the results should be also added in the Figure legend, not only in the text.
ANSWER Thank you for your input. I changed the legend of Figure 1. to:
Figure q. The graph shows the colony-forming units (CFU) per mL, average±SE (n=4) (log scale) of V. anguillarum (a, d), A. veronii (b, e), V. alginolyticus (c, f) in cultures of Tetraselmis sp. (red var.), Tetraselmis sp. (red var., Pappas), Tetraselmis sp. (red var., Kotichi), Tetraselmis sp. (palmella), Tetraselmis marina (var. Messolonghi), Asteromonas gracilis and Amphidinium carterae compared with sterile seawater 25 ppt added Walne's medium (control), over time in light conditions. The microalgae that inhibited the pathogens most efficiently were A. carterae, A. gracilis, Tetraselmis sp. red var., Tetraselmis sp. (palmella), and Tetraselmis sp. (red var., Pappas).
8.Line 148-151, the authors obtained bacterium-free microalgae cultures after antibiotics treatment, and verified by plating on TSA dishes. I don’t think this method can obtain an absolute bacterium-free microalgae culture. An additional bacterial primer-based (such as 16S rRNA gene) PCR method is suggested to verity the bacterium-free status of the tested microalgaes.
ANSWER. We obtained bacterium-free by use of antibiotic mixture as shown in earlier studies. The sensitivity of this method is such that a zero result after plating means that there are less than 20 culturable cells per mL (we plated 50 μL in each plate). This test was done several times before we started the experiment for verification and their results were always negative. The reason we needed these bacterium-free microalgae cultures was to avoid the presence of other bacteria when we inoculated with the pathogens. Even if there were some microalgae cultures with bacteria present, the numbers of bacterial pathogens inoculated in the microalgae cultures at the beginning of each experiment was >10,000 cells /mL. We consider that <20 bacterial cells could not influence the population kinetics of 10,000 cells.
9.The English in the MS need to improve by a native English speaker.
ANSWER. We cannot involve a native English speaker to read through the manuscript, however, the corresponding author has more than 40 publications and we feel confident that the English language should not be an obstacle to the publication of this paper.
10.The starting time in Figure 8 is somewhat chaotic. It is recommended to change the legend to make the image clearer and cleaner.
ANSWER. We changed Fig. 8 to Fig. 6c as suggested by reviewer 1. We hope it is improved now.
11.Line 311, the expression of microbial concentration is wrong. Please check and correct similar problems.
ANSWER. We replaced the expression microbial concentration with the expression concentration of bacteria.
12.IE is for inhibition efficacy or inhibition efficiency? Please check.
ANSWER. It is efficiency, we have made the necessary corrections.
13.It is suggested to illustrate the results of significance test in the legend.
ANSWER. We show in the legends the results of significance test as suggested by the reviewer.
14.If possible, extract the bioactive compounds form the algal culture, and make sure the chemical structures and antibacterial activity quantitatively.
ANSWER. It is our target in a future research to identify in at least some microalgae species the actual compounds that result in antibacterial activity.
15.The format of the some reference is incorrect, for example Ref 12. please recheck.
ANSWER. We have checked again all the references to conform to the format of the journal.
Round 2
Reviewer 1 Report
I am satisfied that the manuscript has been accepted in present form.
Reviewer 2 Report
The revision of the MS has been much improved, and thus can be accepted.